# A Novel GIS-Based Modeling Approach for Evaluating Aquifer Susceptibility to Anthropogenic Contamination

**M. Annie Jenifer [1] and Madan Kumar Jha [2,\*]**

1   VIT School of Agricultural Innovations and Advanced Learning, Vellore Institute of Technology, Vellore 632014, India; anniejenifer@vit.ac.in
2   AgFE Department, Indian Institute of Technology, Kharagpur 721302, India
\*   Correspondence: madan@agfe.iitkgp.ac.in

**Abstract:** Population growth, increasing urbanization and industrialization, mismanagement, and climate change are accountable for the rising depletion and pollution of groundwater worldwide. Consequently, water security, food security, and environmental security are in jeopardy, leading to a severe concern for the sustainable water supply on the Earth. The contamination of groundwater, a complex and hidden resource, is difficult to detect and treat. Therefore, it is essential to evaluate aquifer's propensity for contamination to protect this precious resource. In this paper, a novel approach integrating the GWQI (Groundwater Quality Index), AVI (Aquifer Vulnerability Index), and geospatial modeling is proposed to explore aquifer susceptibility to contamination, applied to an unconfined aquifer. The Groundwater Quality Index (GWQI) was developed by the conventional method and the Analytic Hierarchy Process (AHP), whereas the Aquifer Vulnerability Index (AVI) was developed using a modified DRASTIC model. It was found that the spherical semi-variogram along with simple Kriging is suitable for interpolating concentrations of groundwater quality parameters. Geospatial modeling indicated that the AHP-based GWQI map is more accurate than the conventional method. The integration of the best GWQI and AVI resulted in an Aquifer Susceptibility Index (ASI) map, which revealed that >80% of the study area falls under 'severe' to 'very severe' susceptible zones, while about 20% of the area falls under 'moderate' or 'minimum' susceptible zones. The validation results confirmed that the developed ASI map is reliable. The ASI map can serve as a useful tool for planners and decision makers to devise sustainable aquifer management programs to protect vital groundwater resources from contamination and ensure a safe and reliable water supply under climate change.

**Keywords:** groundwater pollution; aquifer susceptibility index; groundwater quality index; aquifer vulnerability index; geospatial modeling; AHP; modified DRASTIC

## 1. Introduction

A major share of freshwater comes from groundwater, which is a significant and reliable source for human consumption, supplying nearly half of the drinking water demand in the world. Groundwater also supports irrigated agriculture, groundwater-dependent ecosystems, and the socio-economic development of a country [1]. However, about 30% of the world's principal aquifers are under escalating stress due to over-exploitation of groundwater [2]. As of 2010, two-thirds of the total global groundwater is abstracted in Asia, with India, China, Pakistan, Iran, and Bangladesh being major consumers. India ranks first in the world by abstracting 251 km$^3$ of groundwater per year, which is over a quarter of the total groundwater withdrawals in the world [3]. On the other hand, water quality is another important aspect of sustainable water management. The indiscriminate dumping of various solid wastes, improper disposal of liquid wastes from diverse industries, and biomedical wastes from hospitals as well as the excessive use of chemical fertilizers and pesticides in agricultural fields have led to the contamination of groundwater resources in

several parts of the world (both developed and developing nations), which threatens the sustainability of precious groundwater resources and ecosystems [1,4]. The decline in water tables not only results in the depletion of groundwater resources but also induces leakage from contaminated external sources [5], frequently leading to a higher concentration of Arsenic [6], Fluoride [7,8], and other harmful/toxic chemicals in the groundwater. According to SoE [9], groundwater quality is poor in over 200 districts out of 707 districts spread across 29 Indian states, causing serious health risks. Thus, the prevention of groundwater contamination and groundwater depletion is indispensable for the sustainable utilization and management of available groundwater resources.

Given the growing problem of groundwater contamination, it is essential to evaluate the propensity of potential aquifer systems for contamination at a suitable scale using modern tools and techniques. The concept of the Groundwater Quality Index (GWQI) is very useful in overcoming the inherent difficulty in quantifying and expressing water quality [10]. Stigter et al. [11] developed a groundwater quality index for Algarve, Portugal based on multivariate analysis. They reported that the Nitrate contamination and salinization of groundwater in these areas are mainly controlled by Nitrogen from agricultural sources and groundwater recharge. Babiker et al. [12] proposed a modified GIS-based GWQI approach and applied it in the Nasuno basin, Japan. The proposed GQI revealed two gradients of groundwater quality in the basin. Thereafter, this GIS-based water quality index approach was used by several researchers due to its lucidity and easy-to-use quality for elucidating the spatial characteristics of water quality.

The past studies on groundwater quality indexing have used the default interpolation technique available in GIS for integrating the concentration maps of different water quality parameters [12–16] except for a very few studies [17,18]. In the recent past, limited studies have employed a susceptibility index approach to assess the propensity of groundwater for pollution [17,19,20]. Fritch et al. [19] assessed susceptibility of the Paluxy aquifer in North Central Texas and concluded that 27% of the study area was under high susceptibility. Saidi et al. [17] used the susceptibility index approach for the Chebba-Mselloueche aquifer in Tunisia and formulated management criteria for irrigation and drinking water usage in this region. Ncibi et al. [20] evaluated the susceptibility of the Sidi Bouzid basin in Central Tunisia and indicated that 90% of the study area has a high susceptibility to pollution.

Though a few studies have reported the application of a 'susceptibility index' for evaluating aquifer's vulnerability to contamination, none of the studies has adopted an integrated approach of selecting appropriate methods for computing Groundwater Quality Index (GWQI) and Aquifer Vulnerability Index (AVI) in a given hydrogeologic setting. Almost all the past studies on groundwater quality indexing have used the entire available water quality data, i.e., both 'safe' water quality parameters (concentrations within the acceptable limits for drinking) and 'critical' water quality parameters (concentrations close to or more than the permissible limits for drinking). Such a lumped approach for assessing water quality is not technically sound and hence not reliable. Also, it is essential to avoid duplication of water quality parameters in such studies. Therefore, only 'critical' and dissimilar water quality parameters should be considered for the computation of GWQI to ensure reliable and useful results. This study addresses these research gaps by adopting an integrated and technically sound approach for the analysis of aquifer susceptibility. Besides considering only 'critical' and dissimilar water quality parameters, the best-fit interpolation techniques for individual water quality parameters and the most suitable methods for calculating GWQI and AVI have been employed. The validation of the developed Aquifer Susceptibility Index (ASI) map was carried out by using a realistic approach, which is also unique in this study. Thus, the present study is the first of its kind as far as the GIS-based analysis of aquifer susceptibility is concerned.

## 2. Overview Study Area

The study area is located in the Middle Cauvery River Basin of Tamil Nadu, India (Figure 1) with a population of 2,722,290 (2011 Census). It spans from 10°16′ to 11°22′ North

Latitude and 78°15′ to 79°16′ East Longitude encompassing a geographical area of about 4403.83 km². It falls in the sub-tropical climate zone and is comprised of 14 administrative units (locally called 'blocks'). The temperature ranges from 38.5 °C to 29.3 °C. The average annual rainfall in the study area is 820 ± 80 mm, with a majority of the rainfall received from the northeast monsoon. The topography is gently sloping towards the East with a few residual hillocks in the extreme north and south portions of the study area, with elevations 100 m above the mean sea level.

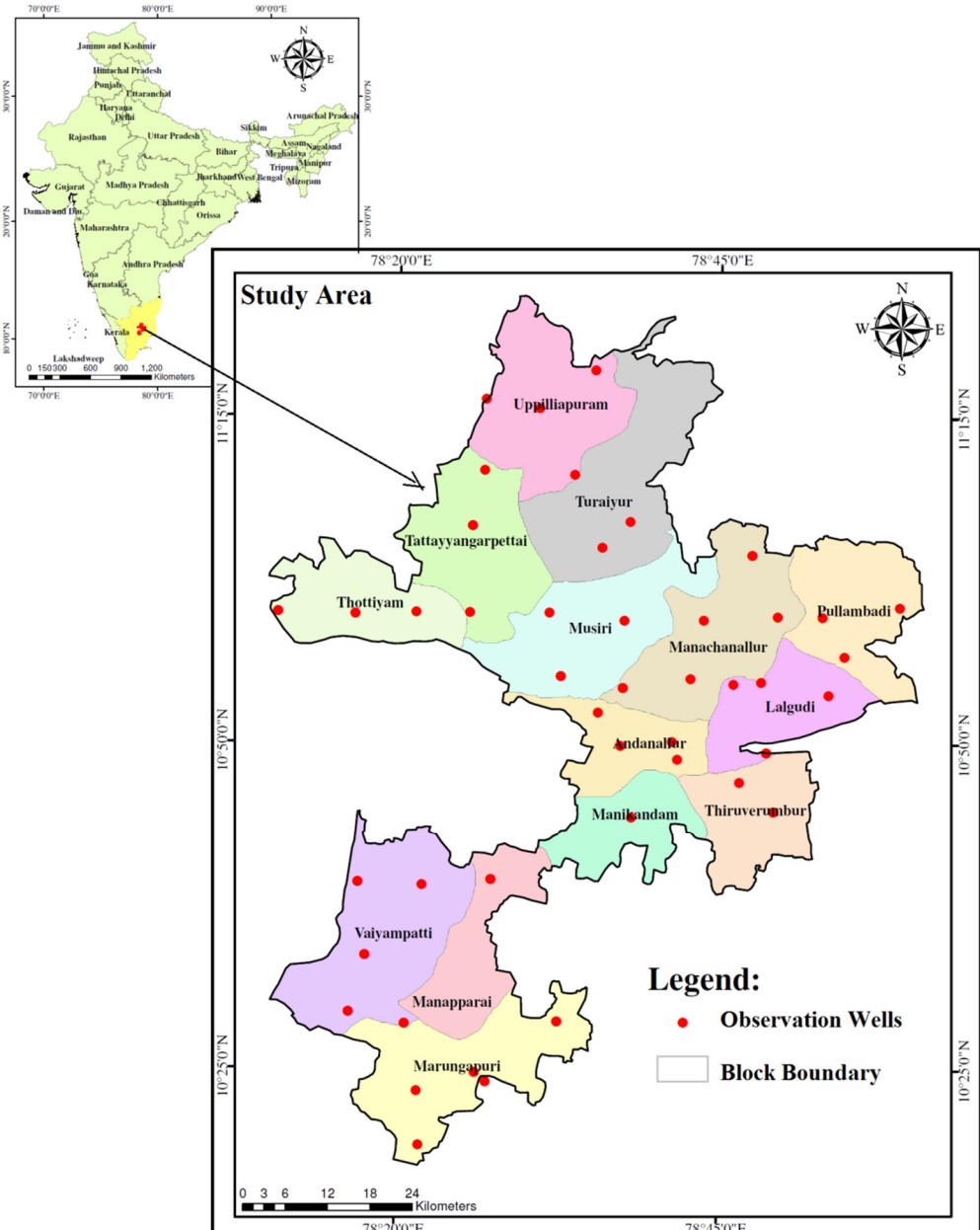

**Figure 1.** Location map of the study area.

The geology of the study area predominantly consists of granite and gneiss in the northern and southern parts of the study area, and alluvium in the middle part of the study area along the Cauvery River. Apart from this, minor compositions of shale, limestone, and charnockite also occur in this region. Groundwater in the study area occurs in unconfined aquifers at depths of 2 to 20 m and in confined aquifers at depths of 20 to 40 m [21]. Approximately 69% of the study area is under irrigated agriculture, while the remaining area is under rainfed cultivation. Irrigated agriculture predominantly depends on groundwater,

a major share of which is from the unconfined aquifers. Therefore, it is of the greatest importance to develop efficient strategies for the sustainable management of groundwater quantity and quality in the study area.

## 3. Methodology

### 3.1. Data Collection

For the assessment of groundwater quality, groundwater quality data for the 46 observation wells were acquired from the Institute of Water Studies, Chennai, Tamil Nadu. The obtained groundwater quality data consisted of 12 major anions and cations. Out of these parameters, seven critical parameters, i.e., Total dissolved solids (TDS), Nitrate ($NO_3^-$-N) (as Nitrogen), Sodium ($Na^+$), Chloride ($Cl^-$), Fluoride ($F^-$), Sulphate ($SO_4^{2-}$), and Total Hardness (TH measured as $CaCO_3$) were selected for groundwater quality indexing. Though increasing the number of groundwater quality parameters improves the prediction of GW quality in the study area, it may also add to the uncertainty and computation power needed. Therefore, the concentration of parameters exceeding the permissible limits for drinking water during the past ten years was identified as 'critical' parameters. These critical parameters are listed by the World Health Organization (WHO, 2004) either under the category of chemically derived contaminants that alter the taste, odor, or appearance of water (TDS, TH, $NO_3^-$-N, $Na^+$, $Cl^-$, $F^-$, $SO_4^{2-}$) or under the category of chemicals that might induce potential health risk ($NO_3^-$-N, and $F^-$). Table 1 summarizes the statistics of seven parameters and the corresponding threshold concentrations, based on the BIS and WHO guidelines.

**Table 1.** Standards used for evaluating potable water quality.

| Sl. No. | Water Quality Parameter | International: WHO [22] | National: BIS (2012) [23] | |
|---|---|---|---|---|
| | | Threshold Value or * Guideline Value | Acceptable Limit | Permissible Limit |
| 1 | TDS (mg/L) | 600 | 500 | 2000 |
| 2 | $Ca^{2+}$ (mg/L) | 100 | 75 | 200 |
| 3 | $Cl^-$ (mg/L) | 250 | 250 | 1000 |
| 4 | * $F^-$ (mg/L) | 1.5 | 1 | 1.5 |
| 5 | * $NO_3^-$ (mg/L) | 50 | 45 | No Relaxation |
| 6 | $SO_4^{2-}$ (mg/L) | 250 | 200 | 400 |
| 7 | TH (mg/L) | 200 | 200 | 600 |

**Note:** *Threshold Value:* The minimum concentration at which taste or odor sensitivity to a particular constituent in water can be perceived; * *Guideline Value*: A numerical value that represents the concentration of the constituents in water that does not result in any significant risk to human health under life-long consumption.

For the assessment of aquifer vulnerability, the groundwater level and litholog data were collected from the Institute for Water Studies, Chennai, Tamil Nadu. These data were used to prepare thematic maps of 'depth to groundwater', 'aquifer media', and 'vadose-zone' media. The soil map was prepared using the soil texture data obtained from the European Digital Archive of Soil Maps. The topographic elevation data, obtained from Shuttle Radar Topographic Mission (SRTM) DEM, were used to prepare the topographic slope map of the study area. The pumping test data collected from the CGWB, Chennai were used to prepare the thematic layers of recharge and hydraulic conductivity of the study area. To delineate a thematic layer of 'lineament density', geomorphology data of the study area were obtained from the Geological Survey of India. Furthermore, the Landsat-8 satellite imagery of 2012 was used to prepare a land use/land cover map of the study area using a supervised classification technique in the ArcGIS environment.

*3.2. Development of the Aquifer Susceptibility Index Map*

Aquifer Susceptibility Index (ASI) provides a typical view of the existing groundwater quality scenario by the integration of the hydrochemical condition (Groundwater Quality Index) and the hydrogeologic condition (Aquifer Vulnerability Index) of the study area. An integrated approach is employed in this study to calculate the susceptibility to aquifer (groundwater) contamination by integrating two indices: Groundwater Quality Index (GQWI) and Aquifer Vulnerability Index (AVI), as illustrated in Figure 2. This approach is applied for the first time in this study by integrating the most accurate GWQI map and AVI map to produce an Aquifer Susceptibility Index (ASI) map on a macro scale.

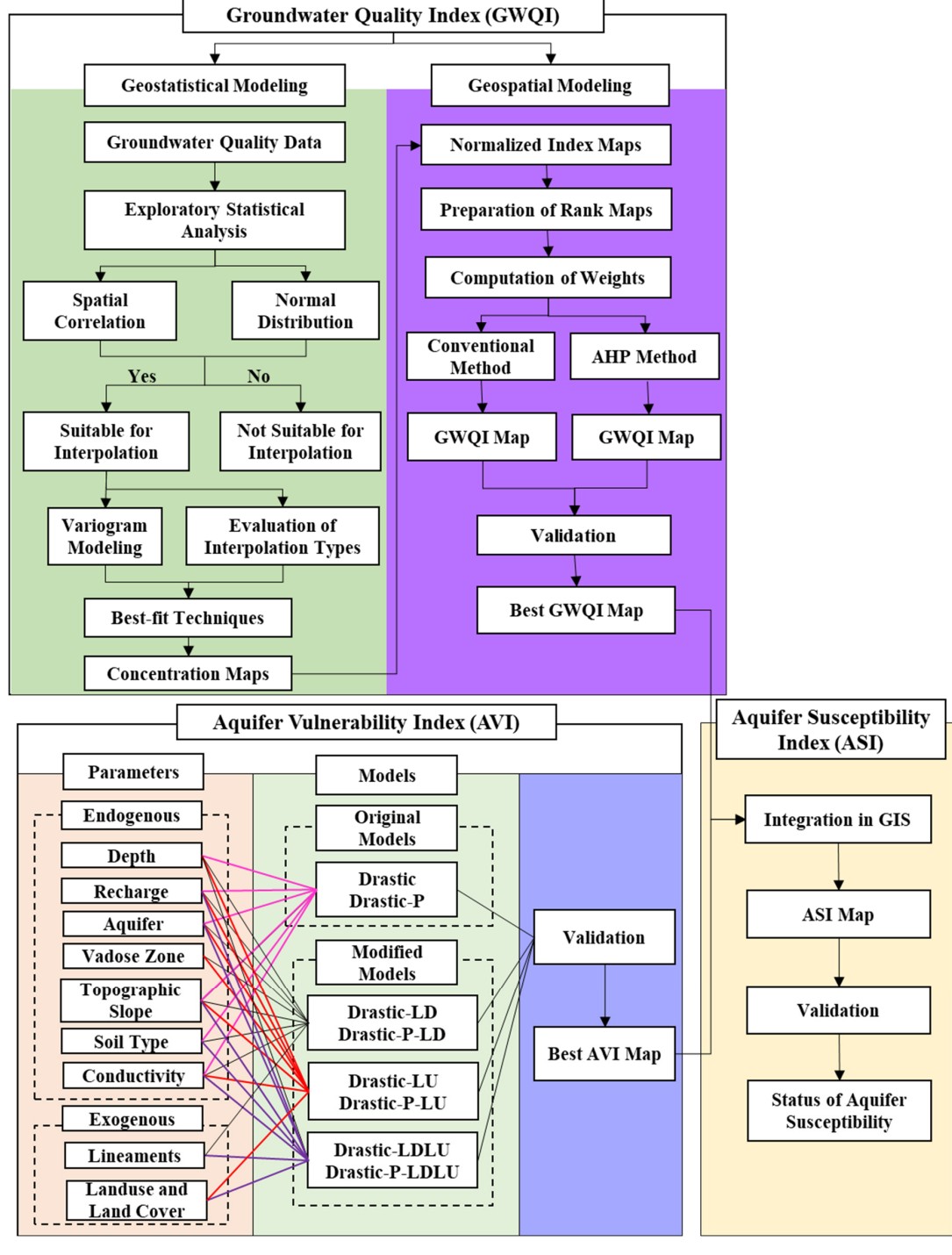

**Figure 2.** Flowchart showing the methodology adopted in this study.

### 3.2.1. Computation of the Groundwater Quality Index

The overall quality of groundwater at a practical scale (e.g., basin or sub-basin scale) is difficult to assess owing to the significant spatial variability of water quality parameters. The water quality index provides a single, dimensionless value representing the overall water quality of a water source based on the presence of several desirable and undesirable constituents in the water [10]. In this study, the GIS-based Groundwater Quality Index (GWQI) model was developed using the groundwater quality data of 46 observation wells, tapping the unconfined aquifer for 2012; the locations of observation wells are shown in Figure 1. The year 2012 was chosen because it represents a 'normal' rainfall year (i.e., annual rainfall within ±10% of its long-period average) in the recent past. Firstly, the available groundwater quality data were evaluated for their suitability for drinking based on the standard guidelines [22]. Thereafter, only salient groundwater quality parameters, having concentrations greater than their permissible limits for drinking, were finally selected for this study. This removal of duplicate parameters is necessary to avoid bias in the analysis. The groundwater quality parameters are shown in Table 1.

For the computation of the GWQI, a systematic and technically sound approach considering geostatistical and geospatial modeling was adopted in this study. The geostatistical modeling involved testing whether the groundwater quality parameters follow a normal distribution, followed by the selection of the best-fit semi-variogram and the interpolation technique for individual parameters to understand the spatial distribution of groundwater quality parameters. The geospatial modeling involved assignment of weights to the groundwater quality parameters by two methods: (i) the conventional method (proposed by Babiker et al. [12]), and (ii) the Analytic Hierarchy Process (AHP) technique, followed by the computation of GWQI based on these methods and the generation of GWQI maps. The procedures of geostatistical modeling and geospatial modeling are succinctly described below.

Geostatistical Modeling

There are a variety of interpolation techniques, which can be classified into two broad groups [24]: (a) deterministic interpolation methods (i.e., linear, polynomial, spline, inverse distance weighted, and natural neighbour), and (b) stochastic interpolation methods (i.e., ordinary Kriging, simple Kriging, universal Kriging, indicator Kriging, disjunctive Kriging, and lognormal Kriging). In this study, Kriging was employed for geostatistical interpolation of water quality parameters due to its salient merits. It reveals the measure of uncertainty or accuracy of the predicted surface and generates a predicted surface from a scattered set of points [25]. In addition, Kriging is generally considered as the Best Linear Unbiased Estimation (BLUE) method of point data [26] since the observation points can be correctly re-estimated. Suitable types of Kriging for each groundwater quality parameter was identified following five steps: (i) spatial correlation analysis; (ii) normal distribution checking; (iii) semi-variogram modeling; (iv) evaluation of interpolation techniques; and (v) cross-validation of semi-variograms and interpolation techniques. The descriptions of these steps can be found in Kitanidis [24].

Geospatial Modeling of Groundwater Quality Parameters

GWQI maps were generated following an automated workflow using the model builder in ArcGIS 10.1, as shown in Figure 3. Concentration maps were generated for each parameter using the best-fit semi-variogram model and interpolation technique identified for individual groundwater quality parameters. The interpolated concentration values were normalized by relating the regional concentration data to the WHO recommended limits for drinking water. For each pixel '$i$' in the concentration map, the Normalized Index values were calculated as follows [12]:

$$NI_i = \frac{C_i{'} - C_i}{C_i{'} + C_i} \qquad (1)$$

where $C_i$ denotes the interpolated concentration value of each parameter and $C_i'$ denotes their recommended limits proposed by WHO for the respective parameter. Using these values, the Normalized Index map for each parameter was developed. The resulting Normalized Index ($NI_i$) values were obtained from Equation (1) in the range of $-1$ to 1. Furthermore, the normalized indices were ranked from 1 to 10 using the following equation (Babiker et al. [12]) that signifies their level of impact on the groundwater quality:

$$r_i = \left(0.5 \times NI_i^2\right) + (4.5 \times NI_i) + 5 \tag{2}$$

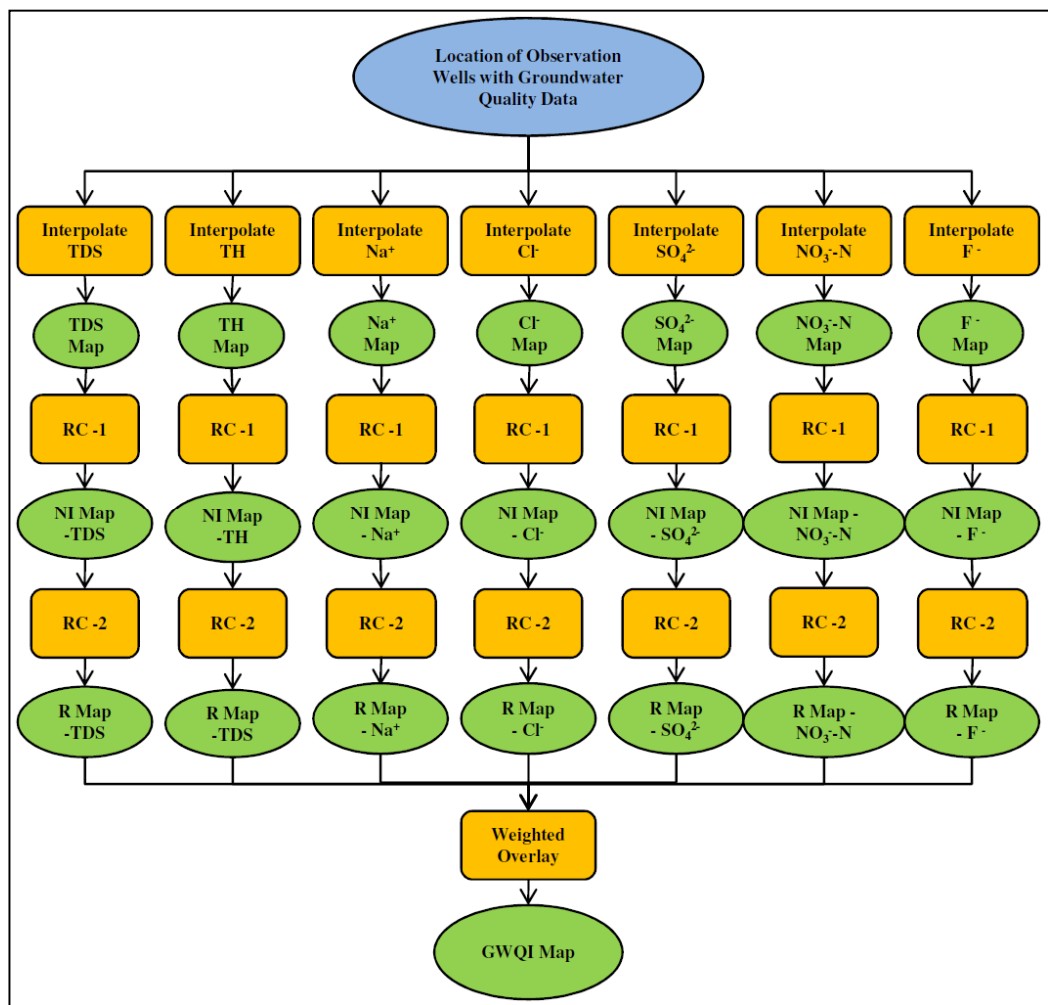

**Figure 3.** Workflow diagram for generating Groundwater Quality Index map. **Note:** RC-1 = Raster Calculator applying Equation (1); RC-2 = Raster Calculator applying Equation (2); NI map = Normalized Index Map; R Map = Rank Map; GWQI Map = Groundwater quality Index Map.

The rank '1' represents the least impact, while '10' represents the highest impact of the parameter on the groundwater quality. Finally, Groundwater Quality Index (GWQI) values were computed from the rank values obtained by Equation (2) using the following equation:

$$GWQI_i = 100 - \left[\frac{r_1 w_1 + r_2 w_2 \ldots + r_n w_n}{n}\right] \tag{3}$$

where '$r$' denotes the rank and '$w$' denotes the weight of a groundwater quality parameter map. In the conventional method, '$w$' is estimated as the mean value of the ranks for individual water quality parameters.

Apart from the conventional method, the Analytic Hierarchy Process (AHP) was also applied to evaluate the groundwater quality in the study area. The AHP method is a widely used Multi-Criteria Decision Analysis (MCDA) technique. In the case of the AHP method, the groundwater quality parameters were weighted based on the number of observation wells in the study area with concentrations exceeding the recommended values over the past ten years. The assigned weights were normalized using the pairwise comparison technique to reduce uncertainties during the weight assessment process. The normalized weights were then added as an attribute to each groundwater quality parameter and all seven parametric maps were integrated using a Weighted Linear Combination (WLC) method to estimate the groundwater quality index values of the study area. The detailed procedure of weight assignment using the AHP technique can be found in Jenifer and Jha [27].

Finally, the GWQI values estimated for all the pixels in the study area were sorted in ascending order and divided into ten classes, with an equal number of pixels in each class. The first class consists of the pixels with the smallest pixel (GWQI) values and the subsequent classes are made up of the pixels with increasing pixel values [28]. This scheme of classification was selected because it does not impose any arbitrary thresholds and also helps in comparing the results obtained by the two methods. The pixels with low GWQI values represent 'poor' quality of groundwater and the pixels with high GWQI values represent 'good' quality of groundwater.

Appraisal of the Predicted GWQI Maps

The validation of GWQI maps is a difficult task and hence it is not a common practice followed by researchers. However, it is necessary to assess the accuracy of the GWQI results. Hence, in this study, the measured concentrations of groundwater quality parameters (groundwater samples collected from the observation wells installed in the unconfined aquifer) were used for the validation of GWQI results obtained by the conventional method and the AHP technique. The observation wells with the water quality parameter concentrations within their acceptable limits for drinking, based on the BIS [23] and WHO [22] guidelines, were categorized as 'Suitable', while the wells with water quality parameter concentrations exceeding the acceptable limits, but within the permissible limits for drinking (Table 1), were categorized as 'Acceptable', while the wells with water quality parameter concentrations greater than the permissible limits for drinking were categorized as 'Unsuitable'. It should be noted that this categorization of observation wells was made considering the fact that the unacceptable concentration of one parameter can render the groundwater unsuitable for drinking. The location map of observation wells was overlaid with the two GWQI maps, and the number of wells falling in each class was identified. Then, two curves were plotted using the number of wells on the *x*-axis and their predicted groundwater quality classes based on the two methods on the *y*-axis.

Sensitivity Analysis of GWQI

The sensitivity analysis refers to the investigation of changes in the results due to some induced variations in the input data. The main intent of this analysis for GWQI is to identify the most sensitive water quality parameter in determining GWQI. The sensitivity of a water quality parameter is expressed in terms of a variation index that is given as [29]:

$$S = \frac{\left| \frac{GWQI_i}{N_i} - \frac{GWQI_i'}{N_i'} \right|}{GWQI_i} \times 100 \tag{4}$$

where $S$ = sensitivity of the GWQI map, $GWQI_i$ = unaltered GWQI of the $i$-th polygon computed using $N_i$ number of parameter maps, and $GWQI_i'$ = altered GWQI of the $i$-th polygon computed using $N_i'$ number of parameter maps.

In the above equation, the term 'Unaltered GWQI' refers to the actual GWQI computed using all the groundwater quality parameters under study, whereas the 'Altered GWQI'

refers to the GWQI computed after eliminating one parameter at a time. Sensitivity (*S*) was estimated by eliminating individual groundwater quality parameters and calculating the change in the groundwater quality index values. Greater variation indicates a higher influence of a particular parameter on the overall groundwater conditions in the study area.

### 3.2.2. Computation of Aquifer Vulnerability Index (AVI)

The vulnerability of aquifer was assessed using the well-known overlay and index model DRASTIC (Aller et al. [30]). However, the original DRASTIC model takes into account only seven endogenous parameters viz., '**D**epth to water', 'net **R**echarge', '**A**quifer media', '**S**oil media', '**T**opographic slope', '**I**mpact of vadose zone media', and 'hydraulic **C**onductivity' (abbreviated as "DRASTIC"). Considering the varied land utilization and weathered rock condition in the study area, the original DRASTIC method/model was modified by adding two exogenous parameters ['land use/land cover' (LU) and 'lineament density' (LD)] and was termed as the modified DRASTIC model (i.e., DRASTIC-P-LDLU). The Aquifer Vulnerability Index (AVI) was calculated as a linear additive combination of weights of the above-mentioned model parameters and the corresponding ratings of their features/classes. The vulnerability maps thus generated, based on the original and modified DRASTIC models, were validated and the most suitable model was selected for computing Aquifer Susceptibility Index (ASI). The detailed methodology for calculating AVI can be found in Jenifer and Jha [31].

Moreover, the most suitable GWQI and AVI models were selected and the two maps were converted into the raster format (grid size of 10 m × 10 m) and superimposed in the ArcGIS environment to compute ASI as follows:

$$ASI = GWQI \times AVI \tag{5}$$

Thereafter, the ASI values were grouped into four classes/zones in the GIS environment to generate a map depicting the susceptibility of groundwater to contamination in the study area.

### 3.3. Validation of Aquifer Susceptibility Index (ASI)

The ASI map developed in this study was validated to find out its accuracy and reliability. To authenticate the obtained results, a novel method of contaminant source identification has been adopted in this study. For this purpose, all the possible sources of groundwater contamination present in the study area were identified. These sources include settlements, solid waste disposal sites, wastewater treatment plants, graveyards, gasoline stations, industries (Iron and steel, leather, paper, oil, chemical, and food), mining areas, and recreation sites. A location map depicting these potential sources of groundwater contamination was prepared using ArcGIS 10.1, which was overlaid onto the ASI map to find out the percentage of contaminant sources in each zone.

## 4. Results and Discussion

### 4.1. Identifying Best-Fit Semi-Variogram Models and Interpolation Techniques

The spatial correlation of the various water quality parameters, examined using Moran's I and Geary's C ratio, are summarized in Table 2, which indicates that the values range from 0.76 to 0.91 and 0.09 to 0.24, respectively. These findings suggest that the groundwater quality parameters exhibit a strong positive correlation and hence are suitable for geostatistical modeling. The results of the Kolmogorov–Smirnov test (Table 3) indicate that the calculated values are less than the critical values for all the groundwater quality parameters. Therefore, the groundwater quality datasets follow a normal distribution, which in turn ensures the applicability of geostatistical modeling.

**Table 2.** Results of spatial correlation analysis for different water quality parameters.

| Sl. No. | Groundwater Quality Parameter | Moran's-I | Geary's-C |
|---------|-------------------------------|-----------|-----------|
| 1 | Sodium | 0.89 | 0.11 |
| 2 | Chloride | 0.83 | 0.13 |
| 3 | Sulphate | 0.88 | 0.12 |
| 4 | Nitrate-Nitrogen | 0.78 | 0.24 |
| 5 | Fluoride | 0.91 | 0.09 |
| 6 | Total Hardness | 0.76 | 0.15 |
| 7 | Total Dissolved Solids | 0.86 | 0.12 |

**Table 3.** Results of the normal distribution test for the groundwater quality parameters.

| Sl. No. | Groundwater Quality Parameter | Value of $K_{(Cal)}$ | Value of $K_{(Critical)}$ | |
|---------|-------------------------------|----------------------|---------------------------|---|
| | | | $\alpha = 0.05$ | $\alpha = 0.01$ |
| 1 | Sodium | 0.103 | 0.273 | 0.227 |
| 2 | Chloride | 0.152 | 0.273 | 0.227 |
| 3 | Sulphate | 0.100 | 0.295 | 0.246 |
| 4 | Nitrate-Nitrogen | 0.175 | 0.295 | 0.246 |
| 5 | Fluoride | 0.190 | 0.418 | 0.314 |
| 6 | Total Hardness | 0.183 | 0.273 | 0.227 |
| 7 | Total Dissolved Solids | 0.183 | 0.273 | 0.227 |

The best-fit semi-variogram model and the interpolation techniques are identified based on three goodness-of-fit criteria, which are summarized in Supplementary Tables S1–S7. It can be seen from the supplementary tables that, in some cases, very little or no variation in the values of goodness-of-fit criteria is found for all four semi-variogram models. In such cases, the simplest semi-variogram model (i.e., spherical) was selected instead of the complicated semi-variogram model (i.e., exponential). Thus, it is evident from Tables S1–S7 that a combination of simple Kriging and the 'spherical' semi-variogram model fits best for all the groundwater quality parameters. However, there are some exceptions, such as the 'Gaussian' semi-variogram and the 'circular' semi-variogram models, found to be the most suitable for $F^-$ and $SO_4^{2-}$ parameters, respectively (Table 4). It is noteworthy that, although the type of semi-variogram model changes for these two groundwater quality parameters, the most suitable interpolation technique (i.e., simple Kriging) remains the same.

In addition, the ratio of the nugget to sill $[C_0/(C_0 + C)]$ was estimated in order to know the spatial dependency of groundwater quality parameters. A relatively small nugget-to-sill ratio indicates a higher accuracy of the geostatistical model in capturing major spatial variation in the groundwater quality [32]. The values of range, nugget, and sill for each parameter are summarized in Table 4. It is obvious from this table that there is 'strong' spatial dependence (ratio $\leq$ 25%) in $Cl^-$, $NO_3^-$-N, TH, and TDS. On the other hand, 'moderate' spatial dependence (ratio = 25–75%) can be seen in $F^-$ and $Na^+$, and 'weak' spatial dependence (ratio $\geq$ 75%) in $SO_4^{2-}$. The 'weak' spatial dependence of sulphate in groundwater can be attributed to the higher spatial variation in sulphate concentration in the zones where land use types are industrial, agricultural, and settlements [33].

**Table 4.** Best-fit semi-variogram models and interpolation techniques for individual groundwater quality parameters.

| Sl. No. | Parameter | Semi-Variogram Model | Interpolation Technique | Range (km) | Nugget ($C_0$) | Partial Sill (C) | $C_0 \times 100/$ $(C + C_0)$ |
|---|---|---|---|---|---|---|---|
| 1 | $Na^+$ | Spherical Model | Simple Kriging | 18.24 | 0.42 | 0.46 | 48% |
| 2 | $Cl^-$ | Spherical Model | Simple Kriging | 17.63 | 0.13 | 0.63 | 17% |
| 3 | $SO_4^{2-}$ | Circular Model | Simple Kriging | 18.47 | 0.94 | 0.12 | 77% |
| 4 | $NO_3^-$-N | Spherical Model | Simple Kriging | 17.08 | 0.17 | 0.54 | 19% |
| 5 | $F^-$ | Gaussian Model | Simple Kriging | 17.72 | 0.81 | 0.91 | 47% |
| 6 | TH | Spherical Model | Simple Kriging | 18.70 | 0.18 | 0.61 | 23% |
| 7 | TDS | Spherical Model | Simple Kriging | 18.8 | 0.23 | 0.86 | 21% |

*4.2. Spatial Distribution of Water Quality Parameter Concentration*

Concentration maps of the groundwater quality parameters were generated using the best-fit semi-variogram models and the best-fit interpolation techniques, as identified in the previous sub-section. The spatial distribution of the concentrations of the seven groundwater quality parameters is illustrated in Figure 4a–g and their brief descriptions are provided below.

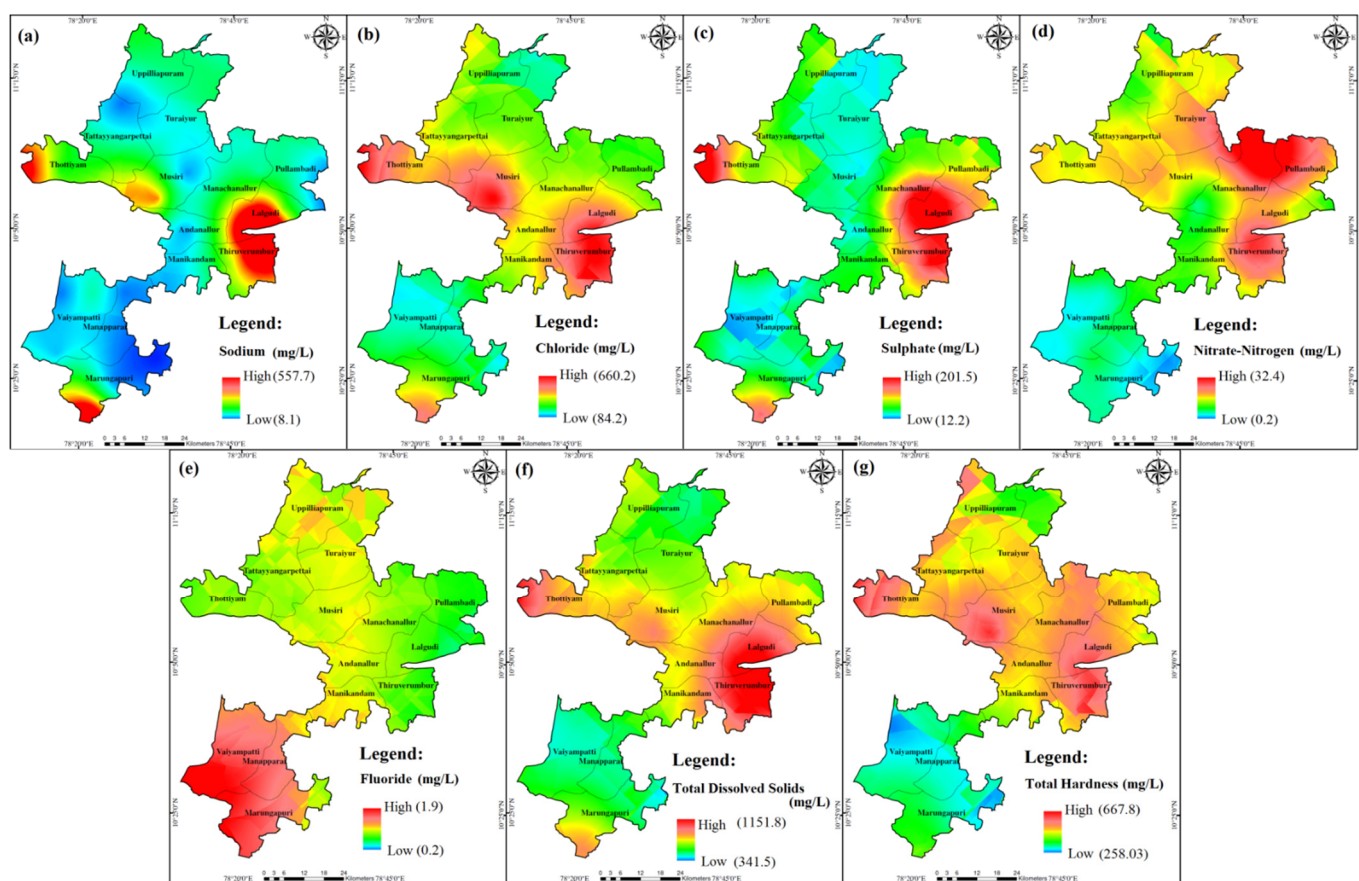

**Figure 4.** (**a**–**g**) Spatial variability of groundwater quality parameters in the study area: (**a**) Sodium; (**b**) Chloride; (**c**) Sulphate; (**d**) Nitrate-Nitrogen; (**e**) Fluoride; (**f**) Total Dissolved Solids; (**g**) Total Hardness.

### 4.2.1. Sodium

The sodium concentration in groundwater Figure 4a varies from 8 to 557 mg/L in the study area. Very high concentration of 500–557 mg/L is apparent in the Thiruverumbur block, owing to numerous metal-processing industries located in this area, where sodium is used as one of the cooling agents. This has also affected the areas around this block viz., Andanallur, Lalgudi, and Manikandam blocks where high sodium concentration is found in the groundwater. The remaining blocks show low to medium sodium concentrations (8–15 mg/L).

### 4.2.2. Chloride

Chloride is generally conservative since it is chemically more stable than other ions and it ranges from 84 to 660 mg/L in groundwater over the study area. Figure 4b shows that the concentration of $Cl^-$ ions in groundwater exceeds the WHO prescribed taste threshold in almost all the blocks except Thuraiyur and Uppilliapuram. The $Cl^-$ concentration in groundwater is generally attributed to saltwater intrusion, the return flow from irrigated land, and the pollutants originating from garbage dumps. Since the study area is considerably far from the sea, the higher $Cl^-$ concentration in groundwater is mainly due to anthropogenic sources. In addition to these sources, the effluents disposed of by the two sugar factories are also responsible partially for the higher $Cl^-$ concentration in groundwater. The use of saline groundwater for irrigation also increases $Cl^-$ concentration in the already saline groundwater. Although there is no immediate health risk due to higher $Cl^-$ concentration, the continuous consumption of drinking water containing high chloride has serious negative health impacts such as colon cancer.

### 4.2.3. Nitrate–Nitrogen

Figure 4c shows that the $NO_3^-$-N concentration in groundwater varies from 1.1 to 32.4 mg/L and exceeds the permissible limit for drinking in almost all the blocks. The primary source of $NO_3^-$-N in groundwater is the agricultural practices followed in the study area (i.e., anthropogenic source). Nitrate can persist in groundwater for decades and accumulate in high levels since more Nitrogen fertilizers are applied to the agricultural fields every year. Other non-agricultural sources of Nitrate include seepage from septic tanks and cesspools, heap of cattle dung, etc.

Moreover, airborne Nitrogen compounds emitted by industries and automobiles are deposited on the land surface as dry particles, which infiltrate during precipitation. Although Nitrate is not a public health threat to adults, its ingestion by infants through drinking water can cause low oxygen levels in the blood—a potentially fatal condition called 'Methemoglobinemia' or 'blue-baby' syndrome. The Nitrogen loading from agricultural sources can be managed by optimizing the quantity and frequency of chemical fertilizers and pesticides with the help of precision farming techniques. Additionally, a paradigm shift from chemical fertilizers to biofertilizers can minimize or avoid nitrate contamination in the groundwater.

### 4.2.4. Fluoride

Fluoride concentration in groundwater in the study area varies from 0.2 to 1.9 mg/L, as illustrated in Figure 4d. Its concentration exceeds the 'guideline value' prescribed by WHO [22], mostly in the southern part of the study area. Fluoride is an important trace element in groundwater, which is generally derived from natural minerals like Appetite, Biotite, Cryolite, Fluorite, etc. The principal anthropogenic sources of $F^-$ in the study area are the combustion of coal and the dumping of fly ash on the land surface by cement industries, mining activities, and the application of Phosphate fertilizers. The burning of coal causes the aerial emission of gases with particulate Fluoride, which percolates during rainfall. Elevated $F^-$ intakes cause more serious effects like skeletal fluorosis (with adverse changes in the bone structure) and crippling skeletal fluorosis, when drinking water contains more than 3 mg and 10 mg of $F^-$ per liter, respectively [7].

### 4.2.5. Sulphate

Sulphate is an essential component contributing to dissolved solids in groundwater. The Sulphate concentration in groundwater is well within the recommended limits in the study area, ranging from 12 to 201 mg/L, as shown in Figure 4e. However, in the Thiruverumbur block and its peripheral regions, its concentration is comparatively high, owing to the use of sulphuric acid and sulphate in metallurgical and glass manufacturing industries located in this region. Also, the Sulphate concentration is increased by the application of gypsum (CaSO$_4$·2H$_2$O) as a soil amendment to improve soil drainage [34].

### 4.2.6. TDS

TDS refers to the measure of dissolved inorganic substances dried at 105 °C and expressed in mg/L units. The spatial distribution of TDS in groundwater, shown in Figure 4f, almost mirrors the sodium and sulphate distribution in the study area. The dissolved solids reduce with a decrease in the groundwater level. TDS values range from about 341 to 1151 mg/L and they exceed the threshold limit for drinking in all the blocks of the study area except Vaiyampatti and some parts of the Manapparai, Marungapuri, and Uppilliapuram blocks. The TDS concentration of groundwater falls within 1000 mg/L in all the blocks of the study area, which suggests that the groundwater is 'freshwater' [35]. However, the TDS of Thiruverumbur and some parts of the Lalgudi and Manikandam blocks exceed 1000 mg/L, which indicates 'brackish water'. Furthermore, although the groundwater with TDS ranging from 1000 to 3000 mg/L is not suitable for domestic purposes, it can be used for irrigation [36].

### 4.2.7. TH

Total hardness generally refers to the dissolved calcium and magnesium in water. The TH values of groundwater range from 258 to 657 mg/L in the study area. However, they are within the 600 mg/L (the permissible limit recommended by BIS [23]) and the water is characterized as 'very hard', according to the classification suggested by Sawyer and McCarty [37]. Figure 4g shows that 'hard water' is mainly found in the central and eastern parts of the study area due to limestone deposits and high dissolved solids in these regions. Although TH as such does not pose serious health effects, it causes limescale formation in plumbing and water heaters, and poor performance of soaps and detergents.

### 4.3. Groundwater Quality Index Maps of the Study Area

Two Groundwater Quality Index (GWQI) maps, using the conventional method and the AHP technique, are shown in Figure 5a,b, respectively. The GWQI, calculated using the conventional method, shows that the groundwater is of 'medium' to 'high' quality since the GWQI values are generally larger (>70). These values were then classified into ten different classes at 10% areal interval. In Figure 5a, there is an apparent gradient in the groundwater quality that decreases with the elevation from north to south since the shallow water table enables faster percolation of contaminants, leading to poor groundwater quality. In addition, in the northern part of the study area, there are relatively less developmental activities (i.e., anthropogenic influence) and hence fewer potential sources of pollutants.

On the other hand, the GWQI map generated using the AHP method, shown in Figure 5b, reveals that the absolute values of GWQI are slightly higher (>90), indicating 'high' quality of groundwater in the study area. Both the GWQI maps (obtained by the conventional and AHP methods) depict similar gradients in the variation of groundwater quality over the study area. However, the AHP method predicts a lesser area (30%) under the 'low' groundwater quality class than the conventional method (42%). In contrast, the area predicted by the AHP method under 'medium' groundwater quality class is greater (52%) than that predicted by the conventional method (43%).

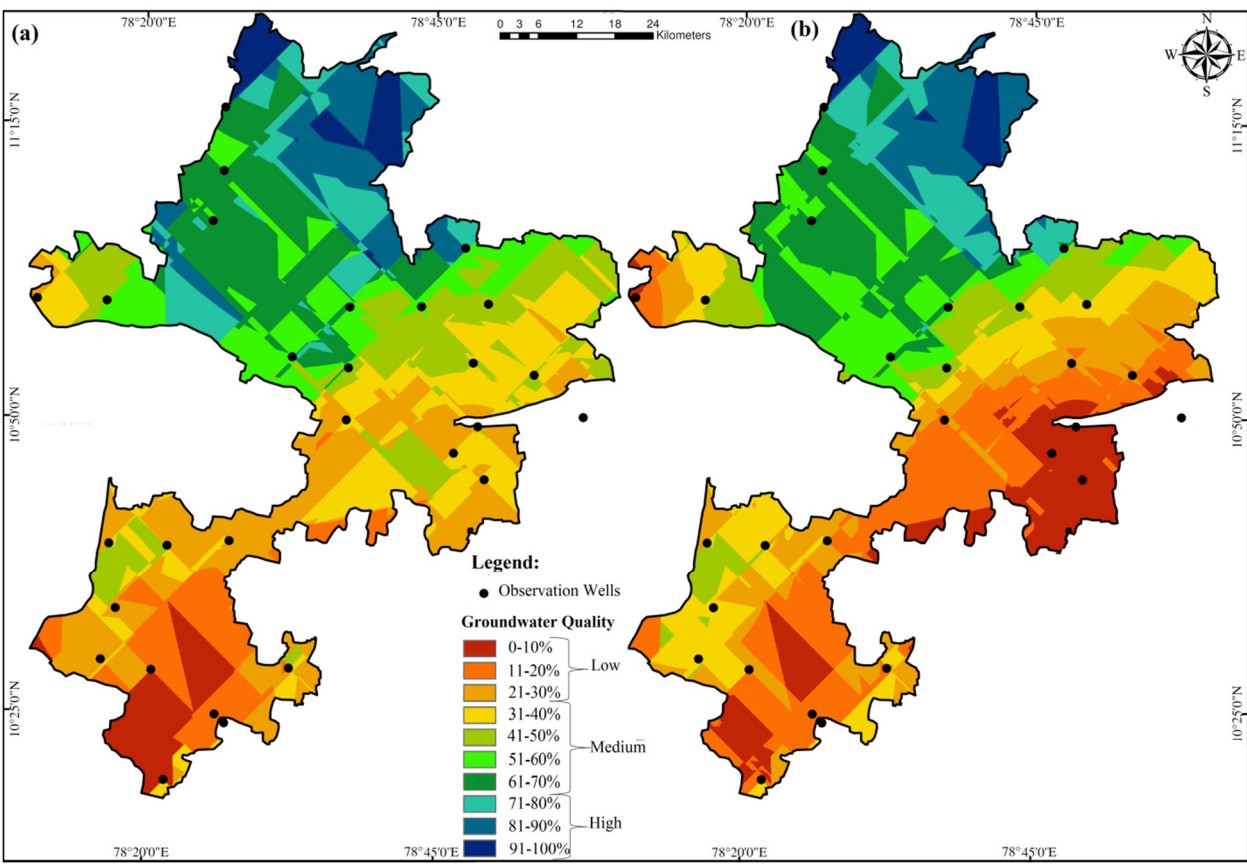

**Figure 5.** (**a**,**b**) Groundwater Quality Index maps based on: (**a**) Conventional method; (**b**) AHP method.

### 4.4. Accuracy of Groundwater Quality Index Maps

The GIS-based GWQI maps were validated with the site-specific concentrations of seven groundwater quality parameters, measured at 46 locations over the study area. It is found that the AHP method shows a high correlation in the case of 'low' and 'medium' groundwater quality classes. However, the 'high' groundwater quality class is more accurately delineated by the conventional method than the AHP method. Further, two (conventional and AHP) curves were fitted to the points as shown in Figure 6, which indicates that the area under the curve for the AHP method is larger than that obtained for the conventional method. Thus, the performance of the AHP method is better than the conventional method in evaluating groundwater quality, thereby indicating that the results of the AHP method are more reliable than that of the conventional method.

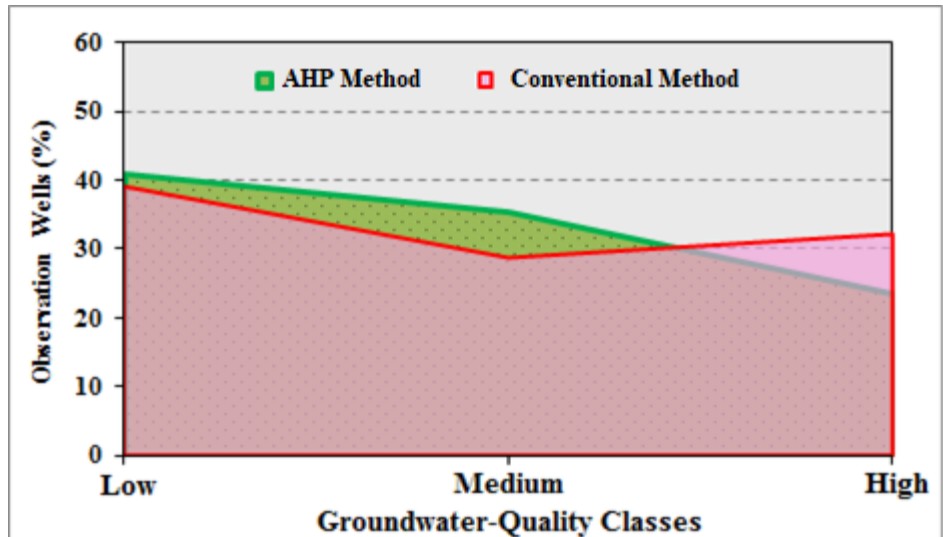

**Figure 6.** Validation of the Groundwater Quality Index maps.

### 4.5. Sensitivity of Groundwater Quality Index Methods

The results of the sensitivity analysis are summarized in Table 5, which indicate that TH is the most influential parameter in the case of the GWQI map based on the conventional method, followed by TDS and $NO_3^--N$, while $SO_4^-$ is the least influential parameter. However, the sensitivity of the GWQI is very small, ranging from 1.7 to 3.0%, as the index was computed using the mean rank values. On the other hand, the results of the sensitivity analysis reveal that the sensitivity of the GWQI based on the AHP method is much less, compared to the conventional method, which could be due to the fact that normalized weights are used in the AHP method for computing GWQI. Thus, the AHP technique predicts more stable GWQI than that predicted by the conventional method, thereby suggesting greater robustness and better technical capability of the AHP technique. Furthermore, the GWQI computed using the AHP technique is highly sensitive to $NO_3^--N$, followed by $Cl^-$ and $F^-$, while it is least sensitive to TDS. Therefore, these water quality parameters (i.e., $NO_3^--N$, $Cl^-$, and $F^-$) must be monitored regularly with the greatest accuracy, compared to other water quality parameters.

**Table 5.** Statistical summary of map removal sensitivity analysis.

| Sl. No. | Groundwater Quality Parameter | Values of GWQI for the Methods | | | |
| | | Conventional Method | | AHP Method | |
| | | Range | Mean | Range | Mean |
|---|---|---|---|---|---|
| 1 | Sodium | 2.63–2.82 | 2.73 | 2.39–2.45 | 2.41 |
| 2 | Chloride | 2.96–4.65 | 3.00 | 2.43–2.45 | 2.44 |
| 3 | Sulphate | 2.30–2.59 | 2.53 | 2.49–2.54 | 2.53 |
| 4 | Nitrate-Nitrogen | 1.67–1.91 | 1.78 | 2.47–2.53 | 2.50 |
| 5 | Fluoride | 2.60–2.81 | 2.68 | 2.71–2.75 | 2.73 |
| 6 | Total Hardness | 1.67–1.88 | 1.70 | 2.51–2.59 | 2.55 |
| 7 | Total Dissolved Solids | 2.20–2.32 | 2.25 | 2.42–2.46 | 2.44 |

### 4.6. Aquifer Vulnerability Index Map

The Aquifer Vulnerability Index map of the study area, generated by the modified DRASTIC model (DRASTIC-P-LDLU), is shown in Figure 7. The AVI values of 99–281 were categorized into four different vulnerability classes, viz., 'very high', 'high', 'moderate',

and 'low' vulnerability of the aquifers. It is evident from Figure 7 that the zone with 'low' vulnerability is in the northern and southern parts of the study area, encompassing about 7% of the area, while the 'very high' vulnerability zone, encompassing 21% of the area, is concentrated around the central portion of the study area. 'Moderate' to 'high' vulnerability zones cover 72% of the total area and are spread throughout the study area. This vulnerability map indicates that the hydrological settings prevalent in the study area can induce surface contaminants to percolate into underlying aquifer systems, especially unconfined aquifers.

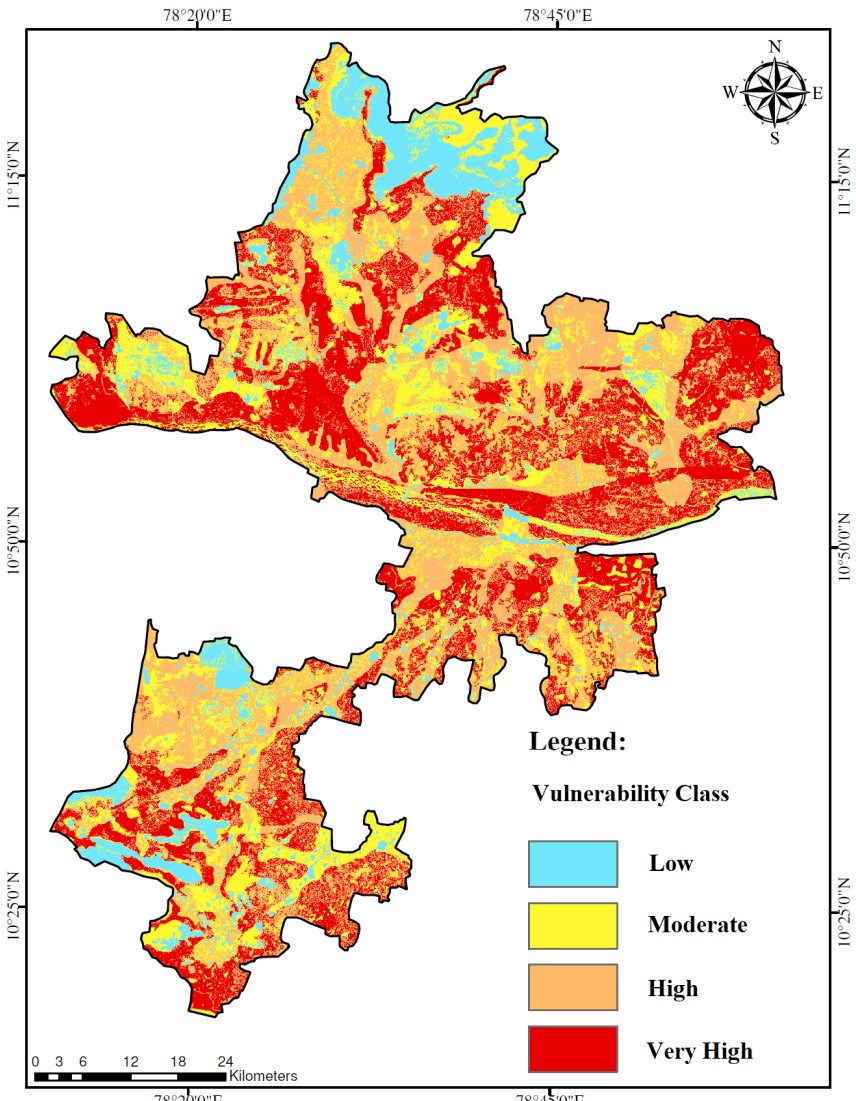

**Figure 7.** Aquifer Vulnerability map of the study area [31].

*4.7. Aquifer Susceptibility to Contamination*

The Aquifer Susceptibility Index (ASI) map of the study area is illustrated in Figure 8. The values of ASI range from 194 to 396 (dimensionless values), based on which the study area can be divided into four classes: (a) minimum (ASI = 194 to 257), (b) moderate (ASI = 257 to 285), (c) severe (ASI = 285 to 309), and (d) very severe (ASI = 309 to 396). The area covered under each class is presented in Table 6; the lowest index values refer to the 'minimum' susceptibility class, which indicates the least susceptibility of the aquifer to contamination and occupies 1.56% of the total area, predominantly in the extreme northern and southern portions of the study area. Two major reasons for the lower susceptibility in these regions could be attributable to geogenic and anthropogenic factors. The geogenic

factors relate to the hydrogeologic setting below the ground, which constitutes rocks made of Anorthosite and Charnockites. These rocks lack primary porosity and therefore minimize the percolation of surface contaminants into the aquifers. The anthropogenic factor in the northern and southern parts of the study area is predominantly agriculture, which reduces the chance of groundwater contamination compared to the regions with industries, mining activities, and urban settlements. The 'moderate' susceptibility class covers 17.61% of the study area and is mostly spread around the periphery of the 'minimum' susceptible zones and partly along the Cauvery River in the middle portion of the study area (Figure 8). The 'severe' susceptibility class covers an area of 55.02% and the 'very severe' susceptibility class encompasses an area of 25.81%, spread throughout the study area and are mainly concentrated in the central portion of the study area. The main factors responsible for the considerably high aquifer susceptibility to contamination in these regions are: (i) flat topography, (ii) an alluvium type of geology that has high permeability and hence faster movement of contaminants, and (iii) highly industrialized and urbanized areas. Thus, these regions host several potential sources of contaminants (Figure 8) originating from different anthropogenic activities.

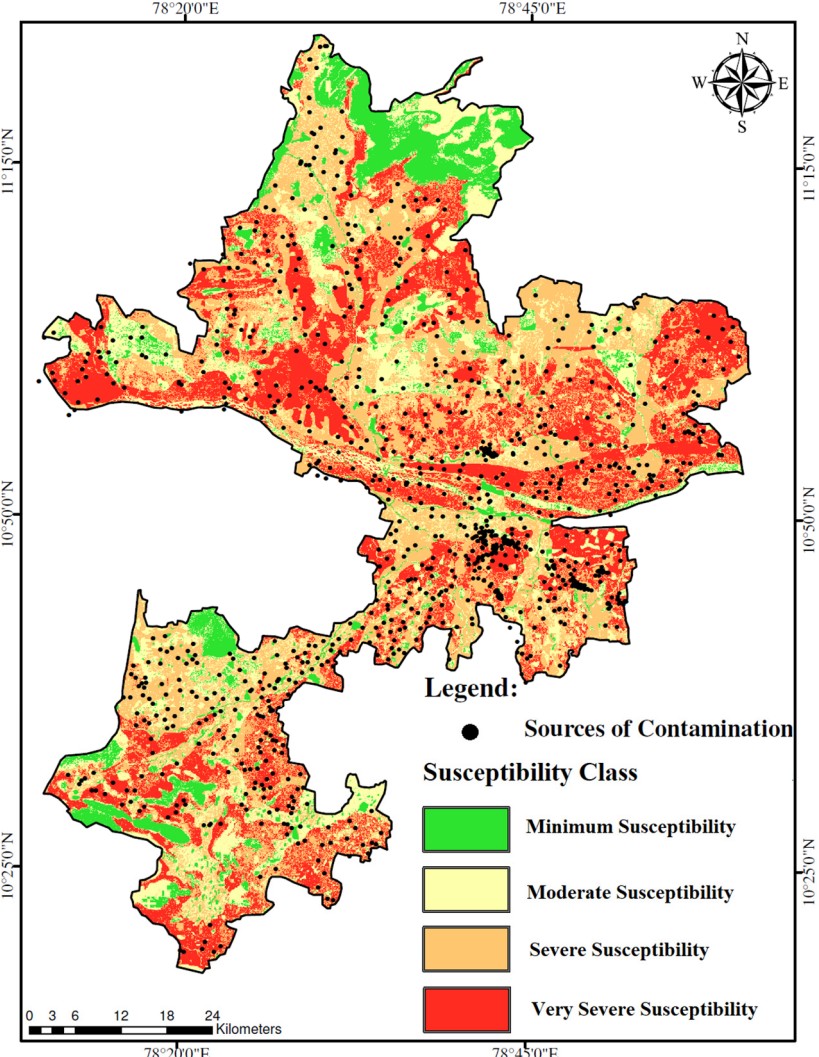

**Figure 8.** Susceptibility status of aquifers to contamination in the study area.

**Table 6.** Classes of aquifer susceptibility and the area under each class.

| Class | Range of Aquifer Susceptibility Index | Susceptibility Level | Area Covered (%) |
|:-----:|:-------------------------------------:|:--------------------:|:----------------:|
| 1 | 194–257 | Minimum | 1.56 |
| 2 | 257–285 | Moderate | 17.61 |
| 3 | 285–309 | Severe | 55.02 |
| 4 | 309–396 | Very Severe | 25.81 |

*4.8. Validation Results of Aquifer Susceptibility Index Map*

The validation of the developed Aquifer Susceptibility Index (ASI) map is shown in Figure 8, wherein the locations of different sources of contaminants over the study area are plotted. It is discernible from this figure that the area under the 'minimum' susceptible zone has the fewest contaminant sources (1.6%), while the 'moderate' zone has 17.6% of the contaminant sources. In contrast, the 'severe' and 'very severe' susceptibility zones host 55% and 26% of the contaminant sources, thereby indicating a greater threat of groundwater contamination in these zones. The current sources of contamination plotted in Figure 8 coincide with the regions with 'severe' and 'very severe' susceptibility of aquifers to contamination. Thus, this confirms that the aquifer susceptibility map developed in this study accurately predicts the susceptibility of aquifers to contamination in the study area.

The aquifer susceptibility map developed in this study can be used by planners, decision-makers, and practitioners to protect vital groundwater resources in the study area in general, and highly susceptible zones in particular. The regions with 'severe' and 'very severe' susceptibility indices are under serious threat and hence the groundwater of these regions must be safeguarded from contamination on a priority basis. In the 'minimum' and 'moderate' susceptibility zones as well, suitable management strategies must be implemented for the protection and management of vital groundwater resources. Appropriate remedial measures such as rainwater harvesting and artificial recharge of groundwater, together with strict regulations for pollution control must be implemented to ensure a sustainable water supply (both quantity and quality) in the study area.

**5. Conclusions**

This study presents a novel approach for assessing aquifer susceptibility to contamination by adopting an integrated and technically sound approach. It was demonstrated through a case study in an unconfined aquifer system underlying the Cauvery River basin of Tamil Nadu, India. In this study, only critical water quality parameters, the best GWQI map and the best AVI map, were employed to develop an Aquifer Susceptibility Index (ASI) map of the study area. Finally, the developed ASI map was validated using a realistic approach.

A combination of simple Kriging and a 'spherical' semi-variogram model was found most suitable for interpolating the groundwater quality parameters. The AHP method was found to be the most reliable for GIS-based groundwater quality evaluation in the study area with weathered hard-rock aquifer systems. The GWQI map generated by the AHP method was then integrated with the best AVI map, generated by the modified DRASTIC model, to produce an ASI map of the study area. The ASI map revealed that more than 80% of the study area falls under 'severe' to 'very severe' susceptible zones, which signify that the groundwater present in these zones is at the highest risk of contamination. On the other hand, about 20% of the study area is under 'moderate' (about 18%) and 'minimum' (about 2%) susceptible zones, which suggest that the groundwater of these zones is at moderate to low risk of contamination. The validation results of the developed ASI map indicate that the aquifer susceptibility to contamination predicted by the proposed integrated approach is accurate and reliable.

The nexus between anthropogenic and geogenic factors provides a composite susceptibility index. The geology (including the vadose zone) of any region is a complex

natural subsurface setting, which cannot be altered by human beings. However, a variety of anthropogenic activities that significantly alter topography and land use/land cover can be minimized, or avoided, to safeguard nature (ecosystems and biodiversity), which in turn can reduce or eradicate human-induced sources of groundwater contamination. Additionally, strict rules and regulations for pollution control/prevention, as well as for hazardous waste disposal and sustainable waste management, are the need of the hour in the study area/region. It is a well-known fact that preventive measures for protecting nature, in general, and groundwater and surface water resources, in particular, are more viable, cost-effective, and successful than remedial measures for several reasons (e.g., natural, technical, and economic constraints). The ASI map developed in this study can serve as a useful tool for planners and decision makers to formulate zone-specific mitigation and adaptation strategies in order to protect scarce groundwater and surface water resources in the study area/region. It is strongly recommended to initiate committed actions in the 'severe' and 'very severe' susceptible zones for protecting precious groundwater resources. On the other hand, it is also essential to maintain the drinkable quality of groundwater in the other two zones ('moderate' and 'minimum' susceptible zones) on a long-term basis. Thus, aquifer susceptibility mapping is instrumental for the planning, decision making, and implementation of policies/scientific strategies for sustainable groundwater management at a large scale. The approach/methodology demonstrated in this study can easily be replicated in diverse hydro-climatic and hydrogeologic regions of the globe.

**Supplementary Materials:** The following supporting information can be downloaded at: https://www.mdpi.com/article/10.3390/su14084538/s1, Table S1: Cross-validation of interpolation techniques for Sodium in groundwater; Table S2: Cross-validation of interpolation techniques for Chloride in groundwater; Table S3: Cross-validation of interpolation techniques for Sulphate in groundwater; Table S4: Cross-validation of interpolation techniques for Nitrate-Nitrogen in groundwater; Table S5: Cross-validation of interpolation techniques for Fluoride in groundwater; Table S6: Cross-validation of interpolation techniques for Total Hardness in groundwater; Table S7: Cross-validation of interpolation techniques for Total Dissolved Solids in groundwater.

**Author Contributions:** M.A.J.: Writing—original draft, Formal analysis, Methodology, Visualization, Resources, Investigation, Validation. M.K.J.: Writing—review & editing, Data curation, Project administration, Supervision, Conceptualization, Funding acquisition. All authors have read and agreed to the published version of the manuscript.

**Funding:** This research received no external funding.

**Institutional Review Board Statement:** Not applicable.

**Informed Consent Statement:** Not applicable.

**Data Availability Statement:** Not applicable.

**Acknowledgments:** The authors are grateful to multiple government agencies such as Survey of India, Geological Survey of India as well as Central Groundwater Board and Institute for Water Studies, Chennai, Tamil Nadu, India, for providing the data necessary for this study.

**Conflicts of Interest:** The authors declare no conflict of interest.

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
