# Peer review of "A Novel GIS-Based Modeling Approach for Evaluating Aquifer Susceptibility to Anthropogenic Contamination"

_sustainability, doi:10.3390/su14084538_

Round 1

Reviewer 1 Report

In the title of the article, it is mentioned that; "A Novel GIS-based Modeling Approach," but I do not see any novelty in this regard.
The maps created in GIS are not available in the article, so it is impossible to comment on the obtained results and their validation.
The article is very descriptive and long. For example, there is no need to provide mathematical formulas for geostatistics analysis and can only be referenced.
In general, the modeling exercise has been conducted nicely using GIS, but I can see the study does not include a suitable conceptual novelty in the current situation.

Reviewer 2 Report

This study presents a novel approach by integrating GWQI, AVI and geospatial modeling to evaluate aquifer susceptibility to contamination. The experiments and the contribution to the community is excellent and meaningful. My main question is that is there some innovation for the methodology in your research. Please explain this and present it in the manuscript. Otherwise, I didn't see any figures in my version of the manuscript.

Reviewer 3 Report

The paper entitled "A Novel GIS-based Modeling Approach for Evaluating Aquifer Susceptibility to Anthropogenic Contamination" is a very interesting document because handled aspects like the identification of pollutants in aquifers. The title is simple but precise.

Nevertheless, it is advisable to change some aspects like as:

  • It is suggested to review deep the English edition.
  • It is recommended that the abstract be simplified to the most relevant results
  • It is recommended that the conclusions be simplified to the most relevant results
  • It is recommended included the Figures, Tables and Maps, because without this information is not possible analysed the result and the manuscript.

Reviewer 4 Report

The authors have presented an interesting work which is definitely of value to Sustainability researchers. The issues I have is mostly with how the paper of written and presented. Here are my suggestions:

1) The 'Introduction' section is too long. May be its best to break the section into 'Introduction' and 'Literature Review'.

2) The 'Data' should be separate from 'Study Area'. May be creating a 'Data and Methods' section would be a better way. 'Development of Aquifer Susceptibility Index' and other related sub-sections can be under the 'Data and Methods' section.

3) There is too much emphasis on the discussion of 'methods' per se rather than the application Indices developed.

4) Not sure if there is a section 5. Because after section 4.8 on 'Validation Results of Aquifer Susceptibility Index Map', it goes directly to section 6. 'Conclusions'. May be it is a typo error.

5) There are too many sub-sections for some of the sections. May be it can be presented in a better way.

Round 2

Reviewer 1 Report

Congratulations, I think It is acceptable according to the applied changes